# How long do floods throughout the millennium remain in the collective memory?

Václav Fanta [1], Miroslav Šálek [1] & Petr Sklenicka [1]

Is there some kind of historical memory and folk wisdom that ensures that a community remembers about very extreme phenomena, such as catastrophic floods, and learns to establish new settlements in safer locations? We tested a unique set of empirical data on 1293 settlements founded in the course of nine centuries, during which time seven extreme floods occurred. For a period of one generation after each flood, new settlements appeared in safer places. However, respect for floods waned in the second generation and new settlements were established closer to the river. We conclude that flood memory depends on living witnesses, and fades away already within two generations. Historical memory is not sufficient to protect human settlements from the consequences of rare catastrophic floods.

[1] Faculty of Environmental Sciences, Czech University of Life Sciences Prague, Kamýcká 129, Praha–Suchdol 165 00, Czech Republic. Correspondence and requests for materials should be addressed to V.F. (email: fanta.vaclav@gmail.com)

t is generally assumed that our predecessors were able to hand information down from generation to generation, and thus to avoid adverse effects of negative events—such as natural catastrophes[1–4]. Collective memory could therefore play a major role in human and communal decision-making, as has been shown by works focused on the ability of humans to learn from technological or natural disasters[5–7]. Other studies however have suggested that this concept works imperfectly, and that learning from history has its limitations[8–12].

People are able to recall memories for decades[13,14]. Diamond[15] underlines the importance of old people for the survival of a community, especially in the past. Before the age of print media, old people acted as keepers of the collective memory of crucial events and issues. Nevertheless, people keep forgetting information. The forgetting curve is logarithmic—the more time that has passed since an event, the weaker are the memories about it[16]. There are many theories on why people forget: spontaneous decay of memory traces, repression of traumatic events, interference with other information, memory noise or loss of the ability to retrieve information stored in the brain[16–18]. As a result, a person or a whole community can forget what was learned in the past.

It is not easy to state for how many years people can reliably remember an item of information, because very few psychological studies about forgetting have dealt with timescales longer than 1 year[19]. Squire[19] found that the answers in a fixed-choice test became random 8 year after an event; however, after 15 years people can recall ~50–60% of information. Hirst et al.[20] reported that even memories of an event as traumatic as the September 11 attacks became inconsistent within 1 year; over the subsequent 9 years, the forgetting curve was approximately constant. Ellis, Semb and Cole[21] showed that students' knowledge declines rapidly 3–7 years after attending a course, though some memories persist for as long as 16 years. On the other hand, memories connected with strong emotions can last for a lifetime[22], as the emotions strongly support memory formation[23].

Studies published so far have usually focused on the period for which a fact or an event is retained in the memory of an individual. Evaluations are usually made on the basis of questionnaires, and not on the basis of the effect that the memory can have on real-life decision-making. More importantly, these studies deal with individuals and not with their interaction inside a group or a community. Only a few studies have dealt with the intergenerational transmission of memory[24, 25]. From the historical perspective, however, it is likely that collective memory and its effect on real-life decisions plays a greater role than the memory of an individual.

Human behaviour after natural disasters (e.g. great floods) is a good model for a study of the collective memory of a community. On the one hand, humans tend to live near the water, because the vicinity of a stable source of the water offers numerous benefits. On the other hand, there is a trade-off between these benefits and a constant threat of flooding. This poses the question—are settlements newly established after floods located in safer sites, or will they preferably be established in close proximity to a water source?

When a flood or a wet climatic period occurred in the past, people often moved their settlements to higher and safer locations, or built new settlements there, or they at least stopped building new houses in dangerous flood zones. This process has been documented in various parts of the world: in Central Europe[2, 26–29], in Great Britain[30–32], in Scandinavia[33], in both Americas[6, 34–36] and in China[37]. The earliest evidence of this process comes from ~4000 BP[33, 37]. Many settlement relocations are reported from the middle ages[29, 35]. Similarly, Collenteur et al.[6] proved that the post-flood population growth in areas affected by the 1993 Mississippi flood in the USA was significantly lower than in unaffected neighbouring areas. They also presented a hypothesis, which was however not tested, that the flood memory would decay over time. On the basis of papers dealing with the persistence of human memory[16, 19–21], we think that flood memory should not start to decay earlier than ~5 years after the flood event.

Previous research on these topics was done predominantly by archaeologists/geographers and by psychologists, mostly presented on the basis of case studies and fragmentary stories. Archaeological/geographical studies have described the relocation of settlements after floods, but have not delved into the duration of flood memory. Psychologists have studied the persistence of human memory, usually by testing how long people could remember information. However, papers studying very long-term memory are rather rare[19]. Extreme floods that occur once in ~100–200 years provide a good opportunity for a natural experiment that can reveal the persistence of historical memory through the behaviour of a community in real situations over several generations.

The aim of our study is to answer the following questions: First, have new settlements been established in the period after major floods in safer sites than before the flood? Second, if so, does this apprehension effect fade away over time, and do new settlements begin to be established closer to the watercourses? Is it possible to determine the length of the flood memory period? Third, can a historical memory effect be observed, i.e., are warnings about rarely occurring great floods passed from generation to generation? Our results indicate that for a period of one generation after each flood, new settlements appeared in safer places. However, respect for floods waned in the second generation and new settlements were established closer to the river. We interpret these results as a consequence of the collective memory, which depends on living witnesses and fades away already within two generations.

## Results

**Vertical distances**. In all four situations (within and outside the Vltava region, one generation before and two generations after flood disasters), the real median vertical distances of new settlements were always less far above the local watercourse than the randomly generated points (virtual settlements), with only two exceptions that will be discussed below (Fig. 1). This confirms that people have historically tended to establish new settlements significantly closer to watercourses than settlements randomly located throughout the landscape would be. The median vertical distance has not changed systematically over the centuries (represented by the sequence of seven flood disasters investigated here) in situations (a), (c) and (d) (all Spearman's correlation coefficients $r_s < 0.53$ and $p > 0.13$), whereas in situation (b), i.e. after flood disasters in the Vltava region, we detected the median vertical distance increasing significantly over the course of the centuries (Spearman's correlation coefficients $r_s = 0.77$ and $p = 0.041$; Fig. 1b). This latter pattern indicates that in the Vltava region, where there is an increased risk of repeated floods, people were well aware of this risk, and took it into account, setting up settlements further above the watercourses in later centuries. An increase in the median distance above watercourses was especially evident in the Vltava region after the 4th flood, in 1501. Indeed, after the flood in 1845, i.e. the last flood in our study, this distance was at the upper limit of the randomly generated virtual settlements.

A comparison of two mixed-effect models analysing the importance of floods as predictors of the decisions of humans on the vertical distance above a watercourse at which new settlements would be set up across generations and through the

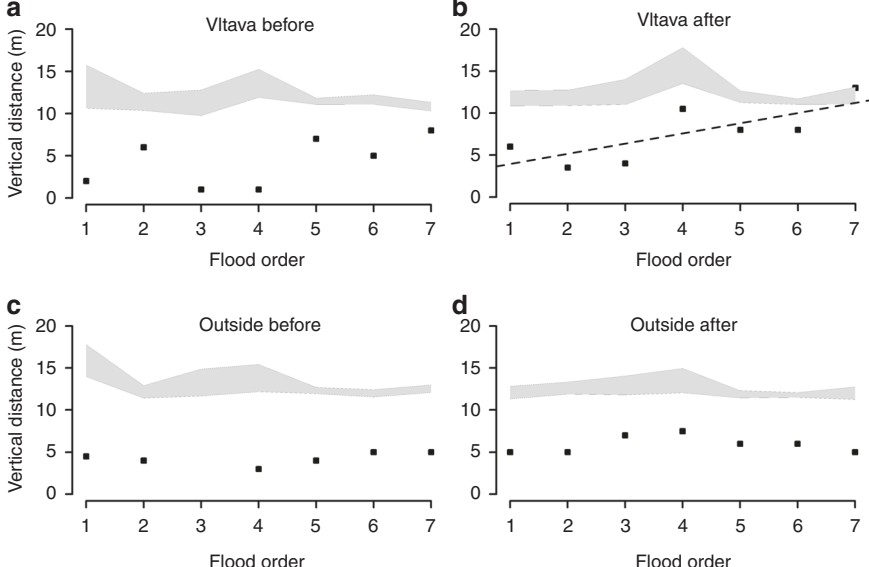

**Fig. 1** Median vertical distances to the nearest watercourses throughout 9 centuries. A comparison between the median vertical distances of newly established settlements (black squares) in the four situations (within and outside the Vltava region up to 25 years before and up to 50 years after flood disasters) and the range of randomly generated points (shaded areas) in the same areas and periods shows that people were strongly attracted towards watercourses. A dashed line refers to a statistically significant trend (see Results). The value before the third flood outside the Vltava was 34 m and lies beyond the presented limit. The variance in vertical distances of randomly generated points may be affected by the different sample size used for the simulation (see Methods section and Supplementary Table 1)

centuries showed that they provided significantly different results ($\chi^2_2 = 8.42$, $p = 0.015$). The generation was either a linear predictor or a polynomial predictor. As the AIC was lower ($\Delta$AIC $= 4.5$) in the model with the polynomial expression of the generation predictor (AIC$_{pol} = 5151.3$) than in the model with the linear expression (AIC$_{lin} = 5155.8$), we selected for interpretation the model with the unimodal response (Table 1a). We therefore suggest that people established new settlements on higher ground in response to flood disasters in the generation immediately following the flood, but later the collective memory faded to some extent in the subsequent (grandson) generation (Fig. 2). This result supports our hypothesis, and indicates that communal memory plays its part, but that it fades away in the second generation. The significant interaction that we observed suggests that the pattern for the Vltava region differed from the pattern for the control region. There was a clear increase in the vertical distance after the disaster, both in the Vltava region and in the control area. In fact, there was a much greater increase in the control area than in the Vltava region, as the vertical distance even before flood was much lower in the control area. It is also evident that in the second generation after the flood, the vertical distance fell to values comparable with those in the period before the flood.

Similarly, we compared two mixed-effect models with the generation stated either as a linear predictor or as a polynomial predictor for the vertical distance above a watercourse from the reference grandson generation to the two subsequent generations (i.e. the third and fourth generations after extreme floods, model B). The models provided similar results ($\Delta$AIC $= 2.9$, $\chi^2_2 = 1.10$, $p = 0.57$) without any significant effects (all $p > 0.22$), suggesting no other significant trend or deflection in the period between the second and fourth generations after the flood (Table 1b; Fig. 2).

**Flood zones**. In the analysis of the human perception of extreme flood situations (which is reflected in the proportions of new settlements established in flood zones), we show that the proportion was almost always lower than the proportion of new

settlements in randomly simulated virtual settlements (Fig. 3). The only exception is the situation before flood 6 (AD 1784), when the real proportion was within the range of randomness.

The results of the GLM analysis of the fixed effects of the periods (before or after the flood), the order in which the floods occurred (1–7) and the interaction of these two predictors on the proportion of real settlements established in the flood zones showed that the proportions of settlements established in the flood zones was not significantly predicted by the period (before versus after the floods; Table 2). However, we detected a significant interaction between the periods and the order in which the floods occurred, indicating that there were different trends in this proportion over the centuries. The proportion of settlements established inside flood zones increased before the floods from the period after the third flood disaster (Spearman rank correlation, $r_s = 0.88$, $p = 0.021$), exemplified by a building boom in high-risk areas particularly in the period since the fifth flood disaster in 1655 (Fig. 3a). In the period after the floods, however, there was no change or only a moderate decline in the proportion of settlements established inside flood zones over the centuries (Spearman rank correlation, $r_s = -0.56$, $p = 0.195$; Fig. 3b). These findings indicate that people continued to be aware of the risks associated with establishing settlements in flood zones for a period of one to two generations after a flood, even during the building boom after 1655. However, it also indicates that population growth in the later centuries (between floods 5 and 7), and the consequent shortage of available low-risk space, led to a need to occupy new areas. This pushed the population into high-risk flood zones.

## Discussion
With just two exceptions, the medians of the real newly established settlements were always located closer to the actual watercourses than the randomly generated settlements (Fig. 1). This may prove that people were always attracted by the presence of water (probably because of their everyday needs), despite the potential risk of floods. It could also explain an issue in

**Table 1 Vertical distances**

| Predictor | Estimate | se | df | $\chi^2$ | *p*-value |
|---|---|---|---|---|---|
| (A) | | | | | |
| Intercept | 1.446 | 0.1127 | | | |
| Longitude | −0.041 | 0.0608 | 1 | 0.459 | 0.498 |
| Latitude | −0.004 | 0.0557 | 1 | 0.011 | 0.916 |
| Vltava (yes vs no) | 0.132 | 0.1348 | 1 | 1.012 | 0.314 |
| Vltava (0): Generation (1) | 3.41 | 2.212 | 4 | 11.29 | 0.024 |
| Vltava (1): Generation (1) | 2.88 | 2.865 | | | |
| Vltava (0): Generation (2) | −4.41 | 2.316 | | | |
| Vltava (1): Generation (2) | −7.23 | 3.04 | | | |
| (B) | | | | | |
| Intercept | 1.571 | 0.1068 | | | |
| Longitude | −0.086 | 0.0948 | 1 | 0.747 | 0.387 |
| Latitude | 0.064 | 0.0788 | 1 | 0.837 | 0.360 |
| Vltava (yes vs no) | −0.029 | 0.2063 | 1 | 0.231 | 0.631 |
| Vltava: Generation | 3.523 | 2.1642 | 2 | 3.015 | 0.222 |

Results of the mixed-effect models that analyse the effects (A) of the first and second generations and (B) the third and fourth generations after flood disasters on the vertical distance of newly established settlements above the nearest watercourse. The reference values are the vertical distances in the period of (A) one generation before the flood event and (B) the second generation after the flood, i.e. one generation before the third and fourth floods. The numeric generation factor (stated in the 2nd order polynomial form) was nested within the Vltava region (within or outside it). Vltava (1) refers to the test area, while Vltava (0) refers to the control area (A)

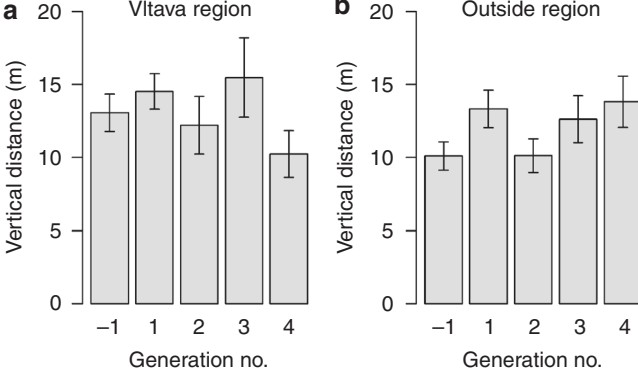

**Fig. 2** Generations before and after the floods. Mean (±standard error) vertical distances of settlements above the water level of the nearest watercourse within **a** and outside **b** the Vltava region in five consecutive generations (25-year periods): −1: before the flood; 1–4: first to fourth generation after the flood

interpreting the results. On the one hand, it is clear that floods are feared (Fig. 2), but on the other hand, construction continued in the flood zones despite the flood risk (Fig. 3). Nevertheless, there is a pattern that bears witness to flood memory, which however fades where great floods are concerned:

In the post-flood periods, there is a retreat to safer locations (Fig. 2, Table 2). After several decades (1–2 generations), however, settlement activity returns to the watercourses (Fig. 2). (All these results were controlled for spatial effect of settlements across the country and were comparable across flood events

which were stated as random effect in the models.) This suggests that there is a limited period of collective memory (a similar result was presented by[8,24,38–40]). We hypothesize that the persistence of flood memory is dependent on the presence in the population of eye-witnesses of the event. After these eye-witnesses die out, the historical memory disappears[41, 42]. The loss of historical memory of extreme events leads in the later centuries to an increase in the proportion of newly established settlements that are located in the flood zones (Fig. 3a). This settlement growth may have been linked, among other things, to the onset of the industrial revolution, and to the resulting increased need for water for industrial processes. This, along with the fading memory of previous extreme floods, may have led the founders of settlements to make a new cost/benefit analysis, and to consider the vicinity of a water source to be more important than the risk of new floods (in the 17th to 19th centuries, ~10–15 % of new settlements were established in flood zones, even after major flood events, Fig. 3b). On the other hand, the vertical distance of settlements established in the test area in the after flood periods was significantly increasing though centuries (Fig. 1b). This result may suggest that the flood memory was passed on to younger generations—which contrasts with our previous findings about the limited duration of flood memory (Fig. 2). Alternatively, newer settlements were established in higher locations (Fig. 1b), but they still remained in the flood zones (Fig. 3b); i.e. the increase in vertical distances does not necessarily indicate the existence of generation memory. Another possible explanation of the increasing vertical distance in time may be the that in the modern period, new settlements were established especially in the uplands and highlands (i.e. in regions with rugged terrain and big vertical differences).

People probably understood the need to build higher above the water level, but lacked information on the precise delineation of the flood zone. Information about the borders of the flood zone that had been acquired empirically in the past was clearly not passed on to future generations. Systematic recording and transmission of detailed flood data only arrived with systematic territorial planning. This developed only in the course of the 20th century, with only sporadic examples in the 19th century[43, 44].

Of course, people's resulting behaviour may not have been due to loss-of-memory alone. For example, the advantages associated with the vicinity of a water source may have played an important role[45, 46]. People may still have remembered the floods and may recognized the associated dangers, but the final decision may have stemmed from the need for a compromise, or for the choice of the lesser of two evils (e.g. safety from floods versus the danger of a potential drought; or the danger of flooding versus the advantages of an adequate supply of water). The decision may have been the result of a trade-off of this kind. The trade-off may have been influenced by the disproportionate probability of favourable and unfavourable impacts—an everyday need for water versus major flooding once in several generations. Apprehension of floods originating from information passed on by earlier generations may not weigh heavily enough against clear present-day advantages, especially if the knowledge has been passed down only verbally, and if disastrous floods occur only once in several generations (cf.[42, 47]).

The relatively short duration of historical memory can be due to several factors: First, the memory of eye-witnesses from the old generation may have weakened by the time they pass information on to their descendants[19–21], or they pass it on imperfectly[12]—because many years have passed since the event. Indeed, the memory is not passed on at all (the old generation has forgotten about it entirely). It has also been observed that repeated (preferably annual) experiences are necessary for proper remembrance[42, 47]. Obviously, this requirement was not fulfilled

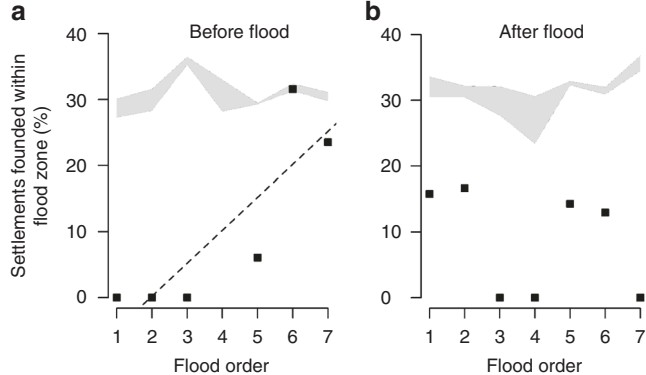

**Fig. 3** Living in the flood zones. A comparison between the proportions (frequencies, %) of new settlements actually established inside flood zones before and after seven historical extreme flood disasters (black squares) and a range of virtual flood events based on randomized simulations (grey area). The dashed line indicates a statistically significant trend (see Results). No data were available before the 4th flood event

**Table 2 Flood zones**

|  | Estimate | se | df | $\chi^2$ | p-value |
|---|---|---|---|---|---|
| Intercept | −3.766 | 3.0619 |  |  |  |
| Longitude | 0.050 | 0.0267 | 1 | 3.685 | 0.055 |
| Latitude | 0.063 | 0.0567 | 1 | 0.460 | 0.498 |
| Vltava | 0.075 | 0.0751 | 1 | 0.949 | 0.330 |
| Order | −0.013 | 0.0163 | 1 | 0.087 | 0.768 |
| Period | −0.327 | 0.1846 | 1 | 0.444 | 0.505 |
| Order: Period | 0.069 | 0.0335 | 1 | 5.391 | 0.020 |

Results of the GLM analysis of the effects of the periods (before or after the flood), the order in which floods occurred (1–7), and the interaction of these factors on the proportion of real settlements established in the flood zones of the Vltava region and the reference area

in the case of extreme floods. Second, the younger generations may not heed the warnings of the ancients, because they seem to them to be ridiculous[48], or the younger generation is not interested in the memories of the older generation. Third, new generations receive the information only by word of mouth, and this weakens the message, and detaches it from the emotions that they would have felt if they had experienced it for themselves[23]. As psychological studies[49] and also neuroscientific studies[50] have pointed out, strong emotions can even strengthen the memory traces via complex biological and chemical processes in the brain. Thus, memories connected with strong emotions (such as personal—living—memories of a dramatic environmental event) are more likely to be remembered, and vice versa: memories received only by listening (i.e. less emotional) are likely to be weaker. The emotional experiences are also more likely to be shared[51, 52]. Although it would be advantageous from the evolutionary point of view to have a better historical memory, our physiology (the decay of memory traces) is a counteracting factor. However, a more detailed analysis of this matter would require a separate study. Social learning, or the ability to acquire information from earlier generations, can also be affected by the cultural environment or by the life (subsistence) strategy of individuals of this type[53–56].

This type of explanation for the duration of historical memory resembles the difference between so-called communicative/lived memory and cultural/distant memory[41, 57, 58]. Living memory is the memory of witnesses who are still alive in the population. Their life stories are known and can still be communicated to the

descendants of the witnesses. Memories recorded in living memory also have an emotional charge[59]. Distant memory, on the other hand, is memory transferred through history textbooks or academic works (the eye-witnesses are no longer alive, and current generations are not emotionally involved). Vansina[41] estimates the line between living memory and distant memory to be not more than three generations, which is somewhat longer than the period indicated in our results (1–2 generations); this difference may be associated with differences in research methods, but the principle remains unchanged. After the death of eye-witnesses, the flood fades from living memory and moves into chronicles and into historical documents. As Pfister[42] noted and as our results have indicated, information written in chronicles and in documents does not affect the real behaviour of ordinary people and communities.

The way of transmission of the flood memory to younger generations can also be an interesting issue. The memory could have been transmitted via personal or written communication[60, 61], cultural artefacts, water marks on the walls of public buildings, religious traditions[42] or in other ways. But this was not the aim of our study and we cede this question to future researchers.

The factors that determine the choice of sites for new settlements may have been affected by local considerations, e.g. by a political decision that failed to take the flood hazard into account, by ownership rights (which in the past often directly affected these sites), or by other natural conditions (for example, in a rugged terrain with narrow valleys, it is impossible to establish a settlement close to a watercourse). We also have to consider that the study presented here is based on current environmental conditions, not on those valid at the time of establishment of the studied settlements; the conditions may have changed somewhat over time in a way that had an effect on decisions about new sites (Pavel Raška, personal communication, 23 June 2017 and 12 April 2018). However, changes in environmental conditions are rarely great enough to have had a significant effect on the results of our study and on the way in which they are interpreted. Other uncertainties in our study may have arisen from the founding dates of settlements: although we compared historical dating with archaeological dating, some unavoidable errors may have remained in the database, and these may have affected the results (e.g. in some examples, the archaeological dating may have been less accurate than expected). However, since most of our data come from the modern period with more precise dating (17th–19th century, see Supplementary Table 1), we think that any errors arising from uncertain dating should be very small.

The lack of statistically significant differences between the test area and the control area (Table 1) was most likely, because many of the major floods assessed in this study affected not only the Vltava basin also other areas of Central Europe as a whole, including major parts of the control area, where there were and still are numerous watercourses with a risk of flooding[62].

Although floods have accompanied humankind since ancient times[62–65], we still often lack sufficient respect for them. It is therefore important to keep reminding ourselves of the risks posed by natural and other disasters, including the causal chain that can lead to the occurrence of floods. Our study overlaps into the fields of social sciences and history: people keep forgetting the danger of natural disasters, and also of social disasters.

To conclude, our study has confirmed the effect of seven well-documented major historical floods on the real behaviour of the communities directly affected by the floods, with reference to the location of new settlements that have been established. We investigated major floods that occurred between the 11th and the 19th century, in order to find out whether these extreme events influenced the height of newly established settlements relative to

the normal water level of the nearest watercourse, and the proportion of new settlements that were established in flood zones. The significant effect of the great floods on both indicators was confirmed on a robust sample of high quality empirical data (1293 settlements established over a period of 8 centuries) reflecting the real behaviour of the community.

The results indicate that for ~25 years after a great flood, new settlements are preferentially established higher above the average nearest watercourse level than before the flood. After that, the locations of new settlements begin to get closer to the watercourses again. A similar effect was revealed through an analysis of the proportion of newly established settlements that were located in the flood zones. The results of the analysis also indicate that the proportion of new settlements that were located within the flood zones grew over the centuries, while this proportion remained roughly constant after the floods.

We interpret our results as a consequence of the collective historical memory. So-called living memory is passed down by living eye-witnesses, and the duration of living memory is apparently conditioned by the life span of the eye-witnesses. The effect of living memory leads to the establishment of new settlements, for a period of one to two generations after the flood, higher up above the watercourses and, to a greater extent, outside the flood zones. However, once the eye-witnesses die out, i.e., after living memory is lost, the community forgets the consequences of such a disaster, and new settlements are established closer to the water again.

People are therefore able to understand complicated processes and situations (which in many cases happened to someone else) and to apply them to new situations. However, this is true only for a limited period of time. The concept of knowledge passed down from generation to generation, especially knowledge of an event in the distant past, is therefore a myth—real data indicate that this is not how we behave, and that information that is not repeated often enough (about once in each generation), fades away from the memory.

Our results imply some important practical considerations. Since it is not safe to rely on folk memory to protect communities from extreme floods, it is all the more important to document extreme floods, and also to bring to people's attention the extreme adverse effects of major flood events. It is essential to keep reminding people of the extent of these events, and to maintain awareness of floods and respect for their impact for a period of decades after the event, especially when no living eye-witnesses remain in the population. It is necessary to teach people about the occurrence of major floods, and about the increasing frequency of these events as a result of climate change. Although flood zones are nowadays relatively well predictable, the exact delineation of flood zones for the purposes of territorial planning are still not universally available. In addition, the risks associated with great floods may be downplayed or simply ignored. The sad result of such attitudes is the sad reality that history keeps repeating itself, even now when reliable knowledge about flood events and about flood prevention is widely available.

## Methods

**Historical floods**. We selected the Vltava river basin in Central Europe as the research area for our study. People have settled there since the early middle ages; the historical floods are well-documented; and the river has a big catchment area. We acquired three types of data: data about historical floods, data about the history of settlements, and data allowing us to measure the relation between settlements and the watercourse. All geographic calculations were done in ArcGIS 10.5.1 (www.esri.com/en-us/arcgis) and QGIS 2.18.15 software (www.qgis.org)[66, 67].

Many historical floods have been recorded in various sources at various sites across the Czech Republic[62], but probably the best data are available for the Vltava river in Prague. We therefore decided to use the Vltava river catchment[68] as a test area, and the rest of the Czech Republic[69] as a control area (Supplementary Fig. 1). We assumed that if a huge historical flood was recorded in Prague (which lies in

the lower part of the catchment), it will also have influenced places situated higher and lower within the catchment area.

The next step was to define the most disastrous floods between 1118 (when the earliest flood recorded in historical documents occurred) and the end of the 19th century (almost no new settlements have been established in the Czech lands since the late 19th century). Extreme floods were chosen for our study because they probably have a greater influence on settlement evolution than small floods. The literature provides various lists of extreme floods in Prague during the second millennium. For our study, we selected seven major floods: 1118, 1342, 1432, 1501, 1655, 1784 and 1845. The criteria for selection were: first, floods greater than a 100-year flood (according to ref. [70]); second, floods with runoff >4000 $m^3 s^{-1}$ (according to ref. [70]; the normal runoff of the Vltava in Prague is ~145 $m^3 s^{-1}$[71]); third, floods recognized as extreme floods by the scientific community[62, 71]. All but one of the selected floods were listed as greater than a 100-year flood[70]. We added the flood of 1501 on the basis of its estimated extreme runoff.

**History of settlements**. There are basically two ways to obtain the date when a settlement was established. The first is historical dating, where there is a record in historical written documents (chronicles, correspondence, narrative sources, official papers, etc.). This type of dating is available for every village or town in the country, but may be unreliable: A settlement may not have been recorded in written sources until many years after its real origin—but the period of the time lag is unknown. The other way is archaeological dating—dating of objects found during archaeological excavations (using various methods, e.g. $^{14}C$, dendrochronology, ceramics, etc.). There is a high probability that the real founding date is captured, but this way of dating is available only for a limited number of places. Another disadvantage of archaeological dating lies in its inaccuracy—e.g. when dating ceramics, we usually know just the century and not the exact year.

In this study, we combined these two approaches. We selected all towns and villages where archaeological research has been performed and where archaeological dating is available with accuracy of a century or better. We then compared the archaeological dating with the historical dating: if the datings coincided (e.g. if the archaeological dating states 13th century and the historical dating states 1258), we used the historical dating. If the archaeological dating was earlier than the historical dating, we used the mid-point of the archaeological dating interval for the calculation (e.g. for first half of 14th century we used year 1325). If the historical dating was earlier, it means that the archaeological dating was insufficiently accurate (artefacts from the oldest phase have not been found); settlements in this category were excluded from our dataset. Moreover, to increase the accuracy of archaeological dating, we worked only with settlements dated with reliable methods (e.g. excavation, profile, trench, field survey). Settlements dated with inaccurate methods (e.g. undocumented research) were excluded from the dataset (we thank archaeologists Jaromír Beneš and Jiří Bumerl for their help with data filtering).

In the 17th century, the state authorities completed the first tax registers (valid for the whole country), in which even small hamlets were described[72, 73]. In modern times, written sources usually record the establishment of a village in the same year as it really was established. We therefore assume that historical dating, for the period since 1600, is accurate enough to be used in our study.

We obtained the historical dating data from the Historical Lexicon of Municipalities[74] and the archaeological dating data from the Archaeological Database of Bohemia[75]. For each flood, we compared the settlements from the test area and from the control area that were established during two time intervals: one generation (25 years) before the flood as a reference, and two to four generations (2–4 × 25 years according to sub-analysis) after the flood to evaluate the response within the first 25-year period (equivalent to one generation) immediately following the flood, and also the response in the subsequent two to four generations (Supplementary Table 1). Our analysis (1 × 25 years before floods plus 2 × 25 years after floods) contains a total of 1314 cases (1293 individual towns and villages). The extended dataset (1 × 25 years before floods plus 4 × 25 years after floods) contains a total of 1669 cases (1637 individual towns and villages). The total numbers of cases are listed in Supplementary Table 1.

**The relation between the settlements and the rivers**. We used two indicators for each settlement: First, vertical distance, which is defined as the height of the settlement above the normal water level of the nearest watercourse (to see how people reflected the normal water level) and second, the proportion of settlements established within flood zones, which expresses whether the settlements were situated inside the flood zones of 100-year floods (to show how aware the local community was of extreme situations, primarily on the basis of empirical experience being handed down about the extent of the flood zones of extreme floods). First, we defined the nearest point on a watercourse from each settlement, using the Near tool in ArcGIS software[66]. We then calculated the elevation of the nearest point and of the settlement itself. The settlements were represented by points placed in the middle of the historical centre of the settlement (for the medieval period, it is not possible to find exact borders of towns/villages, because the oldest available maps with sufficient quality were drawn at the end of the 18th century; we think that a point placed in the middle of the settlement is a good representation). We used elevation data from the DMR 5G digital elevation model (raster data in a 2 m grid) provided by the Czech Land Survey Office[76] and watercourse vector data

provided by the T. G. Masaryk Water Research Institute (TGM WRI)[77]. The Extract Values to Points tool in ArcGIS software was used for the calculation. Finally, we simply subtracted the elevation of the nearest point on the watercourse from the elevation of the settlement. For the second indicator, we used the flood zones of a 100-year flood vector dataset, which was provided by TGM WRI[78]. This dataset is based on current and recent observations. There are two ways in which differences between the current situation and the historical situation could have arisen: horizontal changes (the occurrence of new channels and clogging of old channels) and vertical changes (incision and sedimentation) of the rivers. Horizontal changes have occurred in history (e.g.[79, 80]), but the rivers have usually remained in the current floodplain area during the last millennium. Thus, we assume that the horizontal changes did not affect the extent of the flooded area. The vertical differences between the current position and the early-medieval position of the riverbed could have risen by up to 2 or 3 m in narrow channels[81, 82] (sedimentation caused by deforestation[82, 83], sometimes followed by incision caused by lower intensity of human activities in the last century[82]). As most of the sedimentation occurred during the middle ages[82, 83], we think that our data are not much affected, because most of our data comes from settlements established in the 17th to 19th centuries (Supplementary Table 1). For settlements established during the middle ages, distortion of the extent of the flood zones is theoretically possible. However, we think that the extent of a 100-year flood is much bigger than any changes in the vertical position of the riverbed. To conclude, we assume that the changes from past situations are negligible. Unfortunately, the flood zones of a 100-year flood dataset covers just the main watercourses in the Czech Republic. We therefore had to limit the calculation of the second indicator to towns and villages located in valleys with a defined flood zone (252 settlements met these requirements).

To discover whether people were generally attracted by the proximity of watercourses, we also prepared datasets of random points (virtual settlements) for both indicators. For each period (i.e. 25 years before the flood and 50 years after the flood, 7 floods were investigated, so there was a total of 14 periods), we repeatedly (99 times) generated random points simulating centres of virtual settlements, taking into account the area occupied in previous periods (4-km buffer zones surrounding all settlements established in previous periods were excluded from the area in which the random points were placed). The aim of this step was to generate a range of randomly distributed values in each situation (including the elimination of already populated places), which could be compared with the distribution of real data[84]. We then calculated both indicators (the vertical distance above the normal water level of a watercourse, and the presence of the settlement in a flood zone) for each random point, in order to obtain a range of randomly distributed values in each situation.

**Statistical analysis**. In order to compare real vertical distances with the distances expected by chance (virtual settlements) and to show the general trends across centuries in sums of generations, we analysed the median vertical distances of settlements above a watercourse in the following four situations: within the Vltava region before flood disasters (a), within the Vltava region after flood disasters (b), outside the Vltava region before flood disasters (c) and outside the Vltava region after flood disasters (d) (Fig. 1). In this analysis, we compared the real median vertical distances with the medians for randomly generated points in the areas for particular situations and periods. In this way, we obtained separate results for each (a) to (d) situation. We then tested the temporal trends of the real medians in time-aligned flood disasters (through the centuries), using Spearman's correlation coefficients ($r_s$).

In order to evaluate the general effect of generations in sum of all floods, we selected sets of settlements established one human generation (up to 25 years) before the flood disaster and four generations (4 × 25 years) after the flood, in order to obtain two sets of comparable periods, i.e. one reference period (the first) before the flood and two periods after the flood (for model A) and one reference period (the third period) after the flood and the two following (third and fourth) periods after the flood (for model B). In these models, we tested the significance of the linearity and the unimodality of the response (vertical distances above the nearest watercourses) during two sets of three-generation periods. In the first model (A), we hypothesized that people could have learned from the flood (in the first and second generations after the disaster) and therefore established settlements more safely, i.e. at a greater distance above the watercourses. In this case, we would detect a simple linear response from the reference period to the next two generations. However, if people lost the long-term memory after one generation and initiated a return towards the watercourses (in the second generation), we would detect a more complex unimodal pattern. This pattern would have the greatest distance in the first generation after the flood disaster in comparison with the previous (reference) generation, and also in comparison with the second (grandchild) generation. This pattern may have differed between the higher-risk Vltava region and the other areas. We applied similar treatment to the another model (B), including a set of three successive generations after flood disasters (second, third and fourth) in which we did not expect any significant pattern.

We analysed the predictions using a mixed-effect model (in the *lme4* package) with the response variable representing the vertical distance of the settlement above the water level of the nearest watercourse. Prior to the analysis, the values of the response variable were centred relative to the mean of the respective flood (i.e. first

to seventh), in order to obtain comparable values across periods, and the values were then log-transformed to approach normality. In the models, we tested the fixed effect of three consecutive generations (a three-category predictor, the reference generation prior to a flood and two generations after the flood in model A, and the second, third and fourth generations after a flood in model B). This predictor variable was nested within the region (the Vltava region is compared with the other areas). In both cases (model A: generations −1, 1, 2 as well as model B: generations 3, 4, 5), we compared a model referring to the generations in a linear predictor form (numeric variable generation) with a more complex model referring to the generations in a unimodal arrangement [numeric variable poly(generation, 2)]. In order to select the better candidate model (with a linear effect or with a unimodal effect of the generations), we checked the parsimony of the models, using the Akaike Information Criterion (AIC), and we selected the more parsimonious model with AIC < 2[85]. The results were controlled for longitude and latitude (included as first predictors in the models) to reduce the effect of spatial autocorrelation. The flood event was included as a random factor in order to allow comparisons between particular floods which may vary in average vertical distances.

Finally, we examined how people perceived extreme flood situations, as reflected in the proportions of new settlements established in the flood zones defined by TGM WRI[78] within the selected valleys (regardless of their position in the test area or in the control area) before and after extreme flood disasters. These real proportions were then compared with the proportions calculated from randomly distributed virtual settlements for the respective areas and periods. We applied the generalized linear model (GLM) to test fixed effects of the period (before or after the flood), the numerical order of the flood (1–7), and the interaction of these effects, on the proportion of settlements established inside flood zones. We hypothesized that the proportion of newly established settlements would be lower after the flood event than before it, and that this disproportion would increase over the centuries, due to learning from past experience, at least in the periods after the floods (interaction term). The binary response variable included 1 (present in flood zone) or 0 (outside flood zone). In this model with a binomial error term, we checked for overdispersion by dividing the residual deviance by the residual degrees of freedom. For the purposes of graphic presentation, the proportion (Fig. 3) associated a number of newly established settlements inside the flood zones and outside the flood zones within the selected valleys.

Partial correlations were performed using the Spearman rank correlation coefficient ($r_s$). The presented values indicate the mean ± standard error (se), unless stated otherwise. The models were analysed in R software ver. 3.4.0[86]. The significances are based on likelihood ratio tests, and the level of statistical significance was set at $p = 0.05$.

**Code availability**. The code is available from the authors on request.

## Data availability
We have not performed any investigation on humans or animals. There are no restrictions on data availability. All data are available in the tables or in supplementary materials (Supplementary Data 1–5).

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

## Acknowledgements

We would like to thank Pavel Raška for his fruitful comments, Martin Kuna from the Institute of Archaeology in Prague for providing the archaeological data and Jaromír Beneš and Jiří Bumerl for their assistance with the analysis of archaeological data. We also thank Kateřina Fantová, Jaroslav Janošek and Robin Healey for language assistance. This paper was supported by the Internal Grant Agency of the Faculty of Environmental Sciences, CULS Prague, within the framework of Grant No. 4219013123114 "Use of the Archaeological database of Bohemia in studies of the influence of environmental conditions on cultural landscape evolution in the past". The research also received funding from Czech Science Foundation project GA17-07544S.

## Author contributions

V.F., M.Š. and P.S. designed the research, V.F. collected the data, M.Š. performed the data analysis, and V.F., M.Š. and P.S. wrote the paper.

## Additional information

**Competing interests:** The authors declare no competing interests.

