## [Peer Review File · Nature Communications]

Reviewers' comments:

Reviewer #1 (Remarks to the Author):

The present paper explores how likely it is for a collective memory to be maintained across generations. As the authors point out, the issue is critical, in that the collective memories a community has of the past can shape both collective identity and collective action. The authors address this question in an interesting way: they look at the extent to which a community is willing to settle in the flood zone. They first show that people like to settle near water and then contrast areas that were flooded and those that were not in order to explore how likely resettlement (or more specifically, a minimal height from the flood zone) is one, two, three generations after the flood. Presumably, if the memory of the flood lingers, resettlement (or settlement near the flood zone) should be less likely.

Collective memories can, of course, linger for many generations. We still remember Noah's flood. Such multi-generations memories are, however, usually preserved through societal effort. A community might commemorate the dead each year, or may establish laws that embody the fears the flood engendered. The authors do not consider mnemonic preservation through the use of cultural artifacts. They seem to assume that when it comes to floods, we are dealing with what Assmann called communicative memories — that is, memories conveyed mainly through conversational interactions. Alternatively, they may make no assumptions about how the collective memory is transmitted and preserved. Some may be maintained through cultural artifacts, others through conversational interactions. Future research could separate all this out. What they want to do establish some general estimate of durability across a range of possible means of preservation. This strikes me as a reasonable first step — though I would have liked the authors to be a bit more specific about it.

I am not an expert on floods, so cannot evaluate either the source of information they used nor whether anyone else has done something similar. The authors do cite several studies that involved "flood memory," but argue that this work has not done the systematic analyses they undertake. If so, then the paper makes a substantial contribution.

As a psychologist interested in collective memory, I can, however, speak to the way they discuss the psychological literature and the way they frame their general discussion.

1. I am not sure that the work on how lasting individual memory is germane, at least as they describe it. It is worth pointing out, however, that they are wrong that no one has examined individual

memory retention for more than 16 years. In the experimental literature, Berntsen and colleagues, for instance, have looked at Danes' memory for the weather conditions when Germans invaded and later withdrew from Denmark. Then there are the several studies Bahrick and colleagues that look at, for instance, alumni's memory for their fellow college student, even 50 years after they had graduated. The authors are generally right: There are few studies. But they state the matter in too extreme of terms.

2. But the paper is less about individual memory, but intergenerational transmission of memory. Here I think that authors would be well served to look at the psychological literature a bit more broadly. There is a recent review in *Trends in Cognitive Science* by Hirst, Yamashiro, and Coman. The authors do discuss the work on cultural attractors, but the field is much more diverse than that. In general, the minimal work on intergenerational transmission seems to support the claims the authors are making: that it is quite limited — to perhaps one generation. The convergence in findings should not be viewed as a limitation of the present study. The present study's use of an interesting set of data is *sui generis* and deserves attention.

3. The authors do raise the possibility that people remember the flood clearly, but it does not have the impact several generations down the line that it had if directly experienced or if experienced by one's parents. To me, this possibility still makes the study about memory, but now for the emotional character of the memory rather than the content, if you like. The authors might look at the literature on emotion and individual memory. The emotional content does become less impactful over time. I am not certain that this has been studied, but if it becomes less emotional, it should also be less likely to be transmitted. The relation between transmission of memories and emotions has been studied by Rime.

4. It is unfortunate that the authors could not look at population size, as another and perhaps more sensitive indicator of memory of floods. Some discussion of the limitations of their measures is in order.

5. Finally, it is really up to the editor to decide if *Nature Communications* is the right place for this paper. I can see a good justification for its publication.

Reviewer #2 (Remarks to the Author):

Ref: NCOMMS-18-21089

Title: Floods throughout the millennium – how long do they remain in the collective memory?

Review

There are no competing interests

This paper has a pioneer transdisciplinary approach providing useful material and novel results in the specific field, it demonstrates strong evidence for its conclusions and the data basically sound, but I feel it needs a thorough revision to make it publishable in the Nature Communications. Generally speaking, I find some points where modifications of method, results, discussion and conclusions are suggested.

Please offer a more nuanced discussion.

Please take into consideration the journal style and make the text more readable (e.g. via shorter sentences) for scientific community.

Please develop figures.

Structural changes:

L 155–160: Considering the logic of MS testing the third (and maybe the fourth) post-flooded generations' settlements is reasonable. Please test them and publish the result of the test.

L 170–174: Comparing the deepest point, i.e. the most flood vulnerable part of the studied settlements more reasonable than the altitude of points placed in the middle of the historical centre of the settlement. Please test this indicator too and publish the results.

Statistics:

L 208–211/Fig. 1: "We hypothesized that if people learned from the flood (in the first and second generations after the disaster) by establishing settlements more safely, i.e. with greater distance above the watercourses, we would detect a simple linear response from the reference period to the next two generations." Contrary to this hypothesis, in case of the first and the second floods pre-flood real altitudes in test area (Fig. 1A, B) and in case of the first and the third floods pre-flood real altitudes in control area (Fig. 1C, D) are higher than post-flood altitudes. In case of the 5th flood the difference is probably doesn't significant. Please mention these results and interpret them in discussion and conclusions. This fact will modify discussion too e.g. in L 323: "In the post-flood periods, there is a retreat to safer locations". Thus, the sentence of L 281–282 is not true: "There was a clear increase of the vertical distance after the disaster, both in the Vltava region and in the control area."

L 257–258: In front of the sentence of the result “The median vertical distance has not changed systematically over the centuries (represented by the sequence of seven flood disasters investigated here) in situations (a), (c) and (d) ...” Fig. 1 shows numerous outliers in case of random and real values too. Why?

Figures:

Fig. 1: Please define the trend lines. Value of real in Fig 1C 3rd flood (~33 m) shows something error. Please verify input sides. Different value scale of vertical axes is fallacious! Please correct it. Position of the legend is not fortunate.

Fig. 2: “above the water level” This is not correct. Line 196 mentioned only watercourse. Please correct there the definition and here too. Please move inscriptions of ,control area’ and ,test area’ above the graphs and legends of ,mean’ and ,mean±SE’ to the caption.

Fig. 3: The position of legend is not fortunate. Please move to outside of the graph. Please change ,before’ to ,before floods’ and after too. Please interpret trendlines.

Fig. S1 Legend and caption show duplicated information. Please delete from the legend:

- “ = Vltava r. c. + part of Labe r. c.

- “=rest of the country”

Legend and map show also duplicated information. Please delete from the legend:

- Vltava

- Labe.

In spite of “analysed settlements”, archaeological sites probably more adequate.

Please unify the legend’s information: test area, control area and analysed settlements

Minor comments:

Footnote 1, 2: Beside domestic literature please cite English ones

L 93–94: Too complicated. Traditional word order is more effective.

L 105: Please delete unnecessary information: “of supraregional importance”.

L 117–118: Not coherent sentence.

L 177: In the second occurrence please use the abbreviation of T. G. Masaryk Water Research Institute.

L 178–179: Please offer a more nuanced view on an almost thousand-year long geomorphological transformation of floodplains: “but we assume that any changes from past situations are negligible”.

L 184: Please cite the literature for virtual settlement method?

L 196–200: Please cite the literature for temporal comparisons of vertical distance between watercourses and settlements.

L 208–215: Examples for too long sentences.

L 218–219: The term of 'flood period' hasn't been defined.

L 228: Please introduce AIC.

L 228–230: Please present this step circumstantially: "The results were controlled for longitude and latitude (included as first predictors in the models) to reduce the effect of spatial autocorrelation. The flood event was included as a random factor."

L 232: Repetition, please delete: "defined by the T. G. Masaryk Water Research"

L 290–291: Please move to discussion: "Our results therefore demonstrate that the founders of new settlements more frequently chose sites for new settlements outside flood zones than inside flood zones."

L 308–313: These findings indicate that people continued to be aware of the risks associated with establishing settlements in flood zones for a period of one to two generations after a flood, even during the building boom after 1655. However, it also indicates that population growth in the later centuries (between floods 5 and 7), and the consequent shortage of available low-risk space, led to a need to occupy new areas. This pushed the population into high-risk flood zones.

L 317: Please change "This proves" to "This may prove".

L 318: "Undoubtedly" is too strong. Probably is more adequate.

Reviewer #1 (Remarks to the Author):

The present paper explores how likely it is for a collective memory to be maintained across generations. As the authors point out, the issue is critical, in that the collective memories a community has of the past can shape both collective identity and collective action. The authors address this question in an interesting way: they look at the extent to which a community is willing to settle in the flood zone. They first show that people like to settle near water and then contrast areas that were flooded and those that were not in order to explore how likely resettlement (or more specifically, a minimal height from the flood zone) is one, two, three generations after the flood. Presumably, if the memory of the flood lingers, resettlement (or settlement near the flood zone) should be less likely.

Collective memories can, of course, linger for many generations. We still remember Noah's flood. Such multi-generations memories are, however, usually preserved through societal effort. A community might commemorate the dead each year, or may establish laws that embody the fears the flood engendered. The authors do not consider mnemonic preservation through the use of cultural artifacts. They seem to assume that when it comes to floods, we are dealing with what Assmann called communicative memories — that is, memories conveyed mainly through conversational interactions. Alternatively, they may make no assumptions about how the collective memory is transmitted and preserved. Some may be maintained through cultural artifacts, others through conversational interactions. Future research could separate all this out. What they want to do establish some general estimate of durability across a range of possible means of preservation. This strikes me as a reasonable first step — though I would have liked the authors to be a bit more specific about it.

You are definitely right that the way of transmission of the flood memory can be interesting, but our study was focused on the persistence of flood memory and its impact into the real behaviour of people. The study of the transmission itself was not the aim of our study. We however mentioned this issue in the discussion.

I am not an expert on floods, so cannot evaluate either the source of information they used nor whether anyone else has done something similar. The authors do cite several studies that involved “flood memory,” but argue that this work has not done the systematic analyses they undertake. If so, then the paper makes a substantial contribution.

As a psychologist interested in collective memory, I can, however, speak to the way they discuss the psychological literature and the way they frame their general discussion.

1. I am not sure that the work on how lasting individual memory is germane, at least as they describe it. It is worth pointing out, however, that they are wrong that no one has examined individual memory retention for more the 16 years. In the experimental literature, Berntsen and colleagues, for instance, have looked at Danes memory for the weather conditions when Germans invade and later withdrew from Denmark. Then there are the several studies Bahrick and colleagues that look at, for instance, alumni's memory for their fellow college student, even 50 years after they had graduated. The authors are generally right: There are few studies. But they state the matter in too extreme of terms.

Thank you for recommending the papers by Bahrick *et al.* and Berntsen & Rubin. We have added these citations into our manuscript. We have also toned down the statement about works studying

long-term memory retention. We think that a brief review about the length of individual memory is an important background for flood memory research.

2. But the paper is less about individual memory, but intergenerational transmission of memory. Here I think that authors would be well served to look at the psychological literature a bit more broadly. There is a recent review in Trends in Cognitive Science by Hirst, Yamashiro, and Coman. The authors do discuss the work on cultural attractors, but the field is much more diverse than that. In general, the minimal work on intergenerational transmission seems to support the claims the authors are making: that it is quite limited — to perhaps one generation. The convergence in findings should not be viewed as a limitation of the present study. The present study's use of an interesting set of data is *sui generis* and deserves attention.

Thank you for this point, we slightly modified the introduction section and we added several citations. We think that the mentioned convergence in results is a good sign, as it may indicate the verity of the findings.

3. The authors do raise the possibility that people remember the flood clearly, but it does not have the impact several generations down the line that it had if directly experienced or if experienced by one's parents. To me, this possibility still makes the study about memory, but now for the emotional character of the memory rather than the content, if you like. The authors might look at the literature on emotion and individual memory. The emotional content does become less impactful over time. I am not certain that this has been studied, but if it becomes less emotional, it should also be less likely to be transmitted. The relation between transmission of memories and emotions has been studied by Rime.

We have added a reflection on the importance of emotions in memory persistence into the discussion (5th and 6th paragraphs of the discussion).

4. It is unfortunate that the authors could not look at population size, as another and perhaps more sensitive indicator of memory of floods. Some discussion of the limitations of their measures is in order.

Thank you for this remark. Unfortunately, the earliest (oldest) population data for the whole studied area are from the middle of the 17th century. Since many of our studied settlements were founded in the medieval period, it would not be correct to use roughly approximative data from a distant century. However, because the population size was mainly affected by food availability until the beginning of the industrial revolution (Fanta *et al.* 2018), we think that the population size would not reflect reaction to floods.

5. Finally, it is really up to the editor to decide if Nature Communications is the right place for this paper. I can see a good justification for its publication. **Thank you.**

Cited references:

Fanta, V. *et al.* (2018) 'Equilibrium dynamics of European pre-industrial populations: the evidence of carrying capacity in human agricultural societies', *Proceedings of the Royal Society B: Biological Sciences*, 285(1871). doi: 10.1098/rspb.2017.2500.

Reviewer #2 (Remarks to the Author):

This paper has a pioneer transdisciplinary approach providing useful material and novel results in the specific field, it demonstrates strong evidence for its conclusions and the data basically sound, but I feel it needs a thorough revision to make it publishable in the Nature Communications. Generally speaking, I find some points where modifications of method, results, discussion and conclusions are suggested.

Please offer a more nuanced discussion.

Please take into consideration the journal style and make the text more readable (e.g. via shorter sentences) for scientific community.

Please develop figures.

Thank you, we address all the comments in more detail below.

Structural changes:

L 155–160: Considering the logic of MS testing the third (and maybe the fourth) post-flooded generations' settlements is reasonable. Please test them and publish the result of the test.

We implemented this additional analysis and extended the graphic presentation.

L 170–174: Comparing the deepest point, i.e. the most flood vulnerable part of the studied settlements more reasonable than the altitude of points placed in the middle of the historical centre of the settlement. Please test this indicator too and publish the results.

We understand this argument, but we are afraid that it wouldn't make sense, especially for the following reasons: (1) It is not possible to find exact borders of towns/villages in the time of foundation as the oldest available maps of the studied area are from the end of the 18th century. Therefore, the area and the shape of the built-up area cannot be derived for the first 5 floods in our study (floods in 1118, 1342, 1432, 1501 and 1655). If we don't know the exact shape of a settlement, we are not able to find its deepest point. (2) As we are focused on the persistence of flood memory, we have studied the major floods only. There have been many floods in history, but only a limited number of them were horribly disastrous. Therefore, we want to focus only on events that were threatening to major parts of the towns/villages. We think that the flooding of one or two houses on the periphery is negligible. We are studying the effect of floods on the whole community (not on individuals). Therefore, we think that a point placed in the centre of a settlement is an appropriate representation. (3) The settlements were quite small at the time of their foundation. Thus, we think a point placed in the historical centre properly represents the whole settlement (the historical centre does not differ greatly from the deepest point).

Statistics:

L 208–211/Fig. 1: “We hypothesized that if people learned from the flood (in the first and second generations after the disaster) by establishing settlements more safely, i.e. with greater distance above the watercourses, we would detect a simple linear response from the reference period to the

next two generations.” Contrary to this hypothesis, in case of the first and the second floods pre-flood real altitudes in test area (Fig. 1A, B) and in case of the first and the third floods pre-flood real altitudes in control area (Fig. 1C, D) are higher than post-flood altitudes. In case of the 5th flood the difference is probably doesn’t significant. Please mention these results and interpret them in discussion and conclusions. This fact will modify discussion too e.g. in L 323: “In the post-flood periods, there is a retreat to safer locations”. Thus, the sentence of L 281–282 is not true: “There was a clear increase of the vertical distance after the disaster, both in the Vltava region and in the control area.”

In our opinion this comment indicates a misinterpretation due to our somewhat misleading description in Statistical analysis. First, the hypothesis mentioned here is directly tested using the models summarizing the pattern for particular generations in sum of all floods, for which the results are presented in the outputs of the complex models, and are discussed throughout Fig. 2. On the contrary, Fig. 1 shows only that a) the real distances were almost always systematically lower than the distances expected by chance and b) the trends across the centuries in sum of the distances from the river for the first and second generations increase systematically only in the test area after the flood. However, we cannot use the Fig. 1 to correctly compare individual medians without indicating the variation measure (e.g. boxplots showing quartile overlaps, which we omitted here in order to simplify the interpretation of the whole image). Therefore, we cannot interpret the differences among the medians simply and discuss the results in this light. Our interpretation (the above cited sentence) is based on results of modelling and presenting Fig. 2, which also includes the measure of variation which is necessary for this interpretation. To read the results correctly, we added explanatory sentences into relevant parts of the statistics.

L 257–258: In front of the sentence of the result “The median vertical distance has not changed systematically over the centuries (represented by the sequence of seven flood disasters investigated here) in situations (a), (c) and (d) ...” Fig. 1 shows numerous outliers in case of random and real values too. Why?

We showed that there is only one outlier (Fig. 1c), which may be affected by the different sample sizes available for each category (see Table S1), from which arose also the sample sizes for simulations of virtual distances. We have added this explanation into the description of Fig. 1.

Figures:

Fig. 1: Please define the trend lines. Value of real in Fig 1C 3rd flood (~33 m) shows something error. Please verify input sides. Different value scale of vertical axes is fallacious! Please correct it. Position of the legend is not fortunate.

We have improved the graph: the scales have been unified and the significant trend in Fig. 1b has been defined. The outlier value is correct and refers to an extreme caused by a very limited sample size in this category (see Table S1).

Fig. 2: “above the water level” This is not correct. Line 196 mentioned only watercourse. Please correct there the definition and here too. Please move inscriptions of ‘control area’ and ‘test area’ above the graphs and legends of ‘mean’ and ‘mean±SE’ to the caption.

We have improved the graph according to the recommendations. We measured the vertical distance as “the height of the settlement above the normal water level of the nearest watercourse” (section Materials and Methods, part Measuring the relations between the settlements and the rivers, first sentence). I.e. when referring to the “water level” or the “watercourse”, both terms

mean “the normal water level of the watercourse”. But we agree that the caption to Fig. 2 may be confusing, and therefore we accepted your suggestion and corrected the caption.

Fig. 3: The position of legend is not fortunate. Please move to outside of the graph. Please change ‘before’ to ‘before floods’ and after too. Please interpret trendlines.

We have improved the graph according to the recommendations.

Fig. S1 Legend and caption show duplicated information. Please delete from the legend:

- “ = Vltava r. c. + part of Labe r. c.

- “=rest of the country”

Legend and map show also duplicated information. Please delete from the legend:

- Vltava

- Labe.

Accepted

In spite of “analysed settlements”, archaeological sites probably more adequate.

Because many of our sites had not been archaeologically excavated, we cannot simply call them “archaeological sites”.

Please unify the legend’s information: test area, control area and analysed settlements

Accepted

Minor comments:

Footnote 1, 2: Beside domestic literature please cite English ones

Accepted, we have added new citations

L 93–94: Too complicated. Traditional word order is more effective.

Accepted, word order has been changed

L 105: Please delete unnecessary information: “of supraregional importance”.

Accepted

L 117–118: Not coherent sentence.

Accepted, sentence reformulated

L 177: In the second occurrence please use the abbreviation of T. G. Masaryk Water Research Institute.

Accepted

L 178–179: Please offer a more nuanced view on an almost thousand-year long geomorphological transformation of floodplains: “but we assume that any changes from past situations are negligible”.

Accepted, we have added a more detailed explanation

L 184: Please cite the literature for virtual settlement method?

We have added the citation where this analytical approach is described (Fortin, M.-J. & Dale M.: *Spatial analysis: a guide for ecologists*, Cambridge University Press, 2005).

L 196–200: Please cite the literature for temporal comparisons of vertical distance between water-courses and settlements.

The literature has already been cited in the 6th paragraph of the Introduction (lines 70 – 81).

L 208–215: Examples for too long sentences.

Accepted, sentences reformulated

L 218–219: The term of ‘flood period’ hasn’t been defined.

We have reworded it as the respective flood (i.e. first to seventh) so that the message is understandable.

L 228: Please introduce AIC.

We have extended the sentence with the criterion $AIC < 2$, and we added a citation (Murtaugh, P. A., 2014: In defense of P values. *Ecology* 95, 611–617).

L 228–230: Please present this step circumstantially: “The results were controlled for longitude and latitude (included as first predictors in the models) to reduce the effect of spatial autocorrelation. The flood event was included as a random factor.”

We added a justification in the sentence in the method section (“The flood event was included as a random factor in order to allow comparisons between particular floods which may vary in average vertical distances.”) and highlighted the interpretation in the second paragraph of discussion adding the sentence “All these results were controlled for spatial effect of settlements across the country and were comparable across flood events which were stated as random effect in the models”.

L 232: Repetition, please delete: “defined by the T. G. Masaryk Water Research”

Accepted

L 290–291: Please move to discussion: “Our results therefore demonstrate that the founders of new settlements more frequently chose sites for new settlements outside flood zones than inside flood zones.”

Accepted, sentence deleted

L 308–313: These findings indicate that people continued to be aware of the risks associated with establishing settlements in flood zones for a period of one to two generations after a flood, even during the building boom after 1655. However, it also indicates that population growth in the later centuries (between floods 5 and 7), and the consequent shortage of available low-risk space, led to a need to occupy new areas. This pushed the population into high-risk flood zones.

We are sorry, but you cited a part of our manuscript without specifying what should be improved.

L 317: Please change “This proves” to “This may prove”.

Accepted

L 318: Undoubtedly" is too strong. Probably is more adequate.

Accepted

REVIEWERS' COMMENTS:

Reviewer #1 (Remarks to the Author):

The authors of the study have done a nice job of addressing the concerns of the reviewers. I have only a few, rather picky suggestions to make.

1. Could the authors describe more clearly what the shaded areas of Figure 1 mean?
2. I found some of what was said in the paragraph ending on page 1 and extending to page 2 and the paragraph ending on page 2 and extending to page 3 a bit redundant. A little careful editing should ameliorate this concern.
3. In the discussion on page 10, the authors seem to focus on the possibility that memories have weakened or the older generation does not talk about them. But discussion also requires that the younger generation to be interested. To be sure, they do not want to build or buy a house that is in a potential flood area, but that does not they ask their parents or grandparents about previous floods. Some of the responsibility for poor transmission rests with the younger population's lack of interest, I suspect. Also, again, to burden the authors with more psychology, Bernard Rime has examined how people are more likely to talk about emotional material. His work might be of interest to the authors.

In general, an interesting paper that approaches a topic from what I at least found to be a distinctive angle. By the way, Nature Human Behavior recently published a paper by Candia et al. on the duration of collective memory, using citations as a means of getting at the question.

Reviewer #2 (Remarks to the Author):

Fanta's et al. answers proved 'satisfactorily addressed'.

Reviewer #1 (Remarks to the Author):

The authors of the study have done a nice job of addressing the concerns of the reviewers. I have only a few, rather picky suggestions to make.

1. Could the authors describe more clearly what the shaded areas of Figure 1 mean?

The shaded areas mean the range of randomly generated points. We added this explanation into the figure caption.

2. I found some of what was said in the paragraph ending on page 1 and extending to page 2 and the paragraph ending on page 2 and extending to page 3 a bit redundant. A little careful editing should ameliorate this concern.

We shortened the latter paragraph a little bit.

3. In the discussion on page 10, the authors seem to focus on the possibility that memories have weakened or the older generation does not talk about them. But discussion also requires that the younger generation to be interested. To be sure, they do not want to build or buy a house that is in a potential flood area, but that does not they ask their parents or grandparents about previous floods. Some of the responsibility for poor transmission rests with the younger population's lack of interest, I suspect. Also, again, to burden the authors with more psychology, Bernard Rimé has examined how people are more likely to talk about emotional material. His work might be of interest to the authors.

Thank you for the point about the low interest in the young generation, we added this explanation into the discussion. Thanks for the recommendation of the work of B. Rimé, we added this topic into the discussion.

In general, an interesting paper that approaches a topic from what I at least found to be a distinctive angle. By the way, Nature Human Behavior recently published a paper by Candia et al. on the duration of collective memory, using citations as a means of getting at the question.

Thanks for the suggestion of the paper by Candia et al., we added this citation into our manuscript.

Reviewer #2 (Remarks to the Author):

Fanta's et al. answers proved 'satisfactorily addressed'.